# Micronutrient deficiencies and their co-occurrence among pregnant women in Mbeya region, Tanzania

Geofrey Mchau[1,2]☯, Hope Masanja [1]☯*, Erick Killel[1‡], Kaunara Azizi[3‡], Tedson Lukindo[3], Adam Hancy [4], Stanislaus Henry[1], Heavenlight Paul[2], Abraham Sanga[5], Ramadhani Mwiru[5], Charity Zvandaziva[6], Kudakweshi Chimanya[7], Abela Twinomujuni[1], Ramadhan Noor[8], Ray Masumo[1], Germana Leyna[1,2], Patrick Codjia[5]

1 Department of Community Health and Nutrition, Tanzania Food and Nutrition Centre, Dar es salaam, Tanzania, 2 Department of Epidemiology and Biostatistics, Muhimbili University of Health and Allied Sciences, Dar es salaam, Tanzania, 3 Department of Food Science and Nutrition, Tanzania Food and Nutrition Centre, Dar es salaam, Tanzania, 4 Department of Nutrition Policy and Planning, Tanzania Food and Nutrition Centre, Dar es salaam, Tanzania, 5 Nutrition Section, United Nations Children Fund, Dar es salaam, Tanzania, 6 United Nations Children Fund, Eastern and Southern Africa Regional Office, Nairobi, Kenya, 7 National Dairy Council of USA, Rosemont, IL, United States of America, 8 Nutrition Section, United Nations Children Fund, Addis Ababa, Ethiopia

☯ These authors contributed equally to this work.
‡ EK and KA also contributed equally to this work.
* hopemasanja@gmail.com

**Data Availability Statement:** All data sets underlying this study are freely available at the public repository named Open Science Framework

## Abstract

Micronutrient deficiencies, a global concern affecting vulnerable populations, including pregnant women, pose significant public health challenges. Specifically, micronutrient deficiencies in iron, vitamin A, iodine and folate have been of greatest public health concern among pregnant women. This study aimed to evaluate the co-occurrence of selected multiple micronutrient deficiencies among pregnant women attending Antenatal Care (ANC) in Mbeya, Tanzania. Employing a cross-sectional design, data were gathered from 420 pregnant women aged 15–49 years attending ANC in Mbeya Region. Semi-structured questionnaires captured socio-demographic data, and blood samples were collected for biomarkers assessment. The study used Stata 17.0 to analyze the data. Results revealed that six out of ten pregnant women exhibited at least one form of micronutrient deficiency, with 15.0% experiencing double deficiencies and 2.2% experiencing triple deficiencies. Iron Deficiency Anemia (IDA) was prevalent among 61.7% of anaemic pregnant women. Individual micronutrient deficiency rates were as follows: folate (21.7%), vitamin B12 (9.9%), iron (38.4%), vitamin A (9.8%), with a median Urinary Iodine Concentration (UIC) of 279.4μg/L. Pregnant women with anemia were more likely to have Multiple Micronutrient Deficiencies (MMD) compared to non-anemic counterparts (p-value <0.005). Additionally, those in the second trimester were at a higher risk of MMD than those in the first trimester (p-value <0.005). A higher wealth quantile appeared to protect against MMD (p-value <0.005). The study highlights the co-occurrence of MMD among pregnant women in Tanzania, indicates the necessity for an improved strategy to enhance multiple micronutrients intake through dietary diversity, MMS and fortified foods.

(OSF) through the following link https://osf.io/7ysb9/.

**Funding:** The author(s) received no specific funding for this work.

**Competing interests:** The authors have declared that no competing interests exist.

## Introduction

Micronutrient deficiencies have been a global problem affecting approximately 2 billion people, with pregnant women being among vulnerable groups [1]. Specifically, micronutrient deficiencies in iron, vitamin A, iodine and folate have been of greatest public health concern among pregnant women [1, 2]. The micronutrients deficiencies during pregnancy is due to the significantly increased requirements of energy and nutrients needed to meet physiological demands [3]. Deficiencies of micronutrients are more pronounced in low- and middle-income countries where they frequently occur concurrently, and women often enter into pregnancy while malnourished [4]. Poor micronutrient status during pregnancy has contributed to numerous health problems with anaemia reported as the most common problem. Anaemia caused by iron deficiency in pregnant women accounts for 22% of all maternal deaths worldwide, as well as high rates of preterm deliveries, low birth weight, neonatal mortality, and infant mortality [5, 6]. Like other low-income countries, Tanzania is experiencing 56% anaemia prevalence among pregnant women, with a maternal mortality ratio estimated at 556 deaths per 100,000 live births [7, 8].

Following micronutrient deficiencies and anaemia prevalence reported in various studies, in addition to promote dietary diversity, the WHO recommends the daily use of Iron Folic Acid Supplementation (IFAs) throughout pregnancy. In 2016, the WHO recommended daily intakes for several micronutrients beyond IFA within populations with low dietary intakes or high deficiency prevalence as part of antenatal micronutrient interventions [9]. The need to address micronutrient intake beyond IFA has led to the desire for a single supplement containing all the essential micronutrients for pregnancy. The variation in micronutrient intakes and deficiencies across regions and countries complicates the routine incorporation of Multiple Micronutrient Supplementation (MMS) in ANC services. This variation has also resulted in different micronutrient formulations containing IFA. Due to these complexities, the WHO currently recommends the use of MMS only in the context of research for pregnant women, especially in areas that are more vulnerable [10, 11]. Therefore, this study aimed to provide up-to-date information on the selected micronutrient status among pregnant women aged 15–49 years attending antenatal care (ANC) in Mbeya, a region in the Southern Highlands of Tanzania. Specifically, the study focuses on assessing the prevalence of deficiencies in iron, vitamin A, vitamin B12 and folic acid, as well as iodine insufficiency. Study's findings will inform a comprehensive revision of strategies aimed at enhancing maternal nutrition interventions including leveraging locally available nutrient-dense foods. Additionally, the results will contribute to guide the design of implementation research for MMS among pregnant women, laying the basis for increased access to MMS in Tanzania.

## Materials and methods

### Ethics statement

The study protocol and data collection tools were approved by the National Health Research Ethics Committee (NaTHREC) of the National Institute of Medical Research (NIMR) in Tanzania (ref no SZECH-2439/R.A/V.1/49). Permissions were also obtained from the Mbeya region Resident Medical Officer (RMO), District Medical Officers (DMOs), and health facility authorities. Informed written consent was secured from all participants, and for those under 18, consent was obtained from their parents or guardians. Interviews were conducted confidentially, with all participants informed of their right to withdraw from the study at any time without penalty. The study adhered to the ethical standards of the Helsinki Declaration of 1975.

## Study area

The survey was conducted in Mbeya region, located in the southern highlands of Tanzania. According to the 2022 Population and Housing Census, the region has a population of 2,343,754, out of whom 1,123,828 are male and 1,219,926 are female. Administratively, the region has seven districts councils named Chunya, Mbeya, Kyela, Rungwe, Mbarali, Busokelo and Mbeya City [12].

## Study design

The cross-sectional survey was carried out to assess the micronutrient status of pregnant women attending antenatal clinics in Mbeya region from September–October 2020. The target population of the survey included pregnant women aged 15–49 years within the second trimester of pregnancy and below (<28 weeks of gestation).

## Sampling procedure

A systematic random sampling approach was used in selecting health facilities offering ANC services, as well as pregnant women aged 15–49 years attending antenatal care in health facilities in Mbeya. Probability proportional to size (PPS) was performed to allocate number of facilities per districts for inclusion in the survey. A master list comprising a total of 251 health facilities providing ANC services in Mbeya region was prepared, from which a total of 44 health facilities were randomly selected based on PPS.

   A list of all pregnant women aged 15–49 years attending ANC in the selected health facility in Mbeya was prepared by RCH office for each health facility selected. Systematic random sampling was done by obtaining an accurate and complete list of the pregnant women attending ANC in a particular selected health facility. The sampling interval K was then computed to select participants. Selection was done systematically with replacement after every $K^{th}$ pregnant woman to participate in the survey.

## Sample size determination

Total of 420 respondents were involved in the survey. A sample size was determined based on the proportion of anaemic pregnant women (25.3%) in the Mbeya region as reported in the Tanzania DHS 2015/16. The sample size was calculated at a 95% level of significance, with a 5% margin of error, and a design effect of 1.5.

## Study population

All pregnant women aged between 15 to 49 years with gestational age less than 28 weeks attending ANC clinics in Mbeya were invited to participate in the study. Gestation age of study participants was read from each participant ANC card, since the catchment area was ANC clinics. Of the 574 eligible women, 420 consented, yielding a response rate of 73%. Participants unable to provide informed consent due to illness or medication use, and those who declined participation were excluded from the study.

## Data collection

A face-to-face interview was conducted among pregnant mothers through the administration of a semi-structured questionnaire by the use of the Open Data Kit (ODK) to collect data on socio-demographics, maternal nutrition and pregnancy. Mid- Upper Arm Circumference (MUAC) measurements were taken using MUAC tapes, followed by blood and urine sample collection for laboratory analyses for those who further consented to clinical assessment. In

clinical assessment, the targeted micronutrient biomarker indicators involved were anaemia, vitamin A, iron, vitamin B12, red blood cell folate and urinary iodine.

## Wealth quantile

Households were assigned scores based on the ownership of various consumer goods (such as televisions, bicycles, or cars) and housing characteristics (including the source of drinking water, toilet facilities, and flooring materials). These scores were calculated using principal component analysis. The wealth quintiles were established by assigning each household's score to its members, ranking them by score, and then dividing the population into five equal categories, each representing 20% of the population [13].

## Sample collection

In each health facility, a temporary laboratory was set up for sample collection and field testing. A trained nurse collected blood samples through vein puncture from consented participants. Blood samples were taken into ethylenediaminetetraacetic acid (EDTA) and non-anticoagulated whole blood vacutainers (Becton Dickenson, NJ, USA). Approximately Venous blood sample (6mL) was collected on each vacutainer and protected from light. Whole-blood vacutainers were maintained at 4–8˚C for less than 2 hours before transport to the temporary laboratories.

At the temporary laboratories, complete blood count (CBC) determination was performed by using a Sysmex XP-300 automated hematology analyzer (Sysmex Corporation, Kobe, Japan). The whole blood sample for red blood cell folate, was treated with 1% ascorbic acid, and the hemolysate was stored at −80˚C until analyzed. Serum were separated and aliquoted into vials and stored at −80˚C until analysis.

## Laboratory sample analysis

**RBC folate measurement.** Whole blood and serum folate concentration was determined by folate microbiological assay, as previously described by O'Broin and his colleague [14], using standardized kits and protocol from the US Centres for Disease Control and Prevention, Atlanta GA. The method uses chloramphenicol-resistant organism, *Lactobacillus rhamnosus* (ATCC 27773; American Type Culture Collection, Manassas, USA) as a test organism and 5-methyltetrahydrofolate (Sigma-Aldrich) as the calibrator. Since we used 5-methyl tetrahydrofolate as a calibrator from CDC USA which gives lower RBC folate concentrations than folic acid calibrator used by WHO assay, we used adjusted cutoff of >748 nmol/L as established previously [15, 16]. A RBC folate concentration <748 nmol/L was used to indicate folate insufficiency. Quality control was ensured with a coefficient of variation <15% and a detection limit <110 nmol/L.

**Vitamin B12 assessment.** Vitamin B12 levels were measured using the Roche Elecsys Cobas e411 immunoassay analyzer (Roche Diagnostics GmbH, Germany), which uses the competitive electrochemiluminescence immunoassay "ECLIA" test principle. Quality-control samples with low and high ranges of vitamin B12 were analyzed along with the samples. Vitamin B12 deficiency was defined as serum vitamin B12 level of ≤148 pmol/L [17]. The reference method's coefficient of variation was <10%, and the assay detection limit was <100 nmol/L.

**Assessment of serum Ferritin.** Assessment of Serum Ferritin was performed with Roche Cobas Integra 400 Plus analyzer (Roche Diagnostics GmbH, German). Hemoglobin levels <11.0 and <10.5 g/dL were used to characterize anaemia for pregnant women in the first and second trimester respectively [18]. Hemoglobin levels <7.0 g/dL was used to characterize severe anaemia. Serum Ferritin < 15.0 g/L was used to identify iron deficiency. Hb level <11.0 g/dL

with Serum Ferritin <15.0 µg/L was used to define iron deficiency anaemia [19]. Adjustment for Ferritin concentration was made by regression approach as applied in the BRINDA project [20]. The method had a coefficient of variation <10% and a detection limit <10 nmol/L.

**Assessment of serum retinol.** The concentrations of retinol in serum were determined by high-performance liquid chromatography (HPLC) System (Shimadzu, Duisburg, Germany). Briefly, serum concentrations of vitamin A (retinol) were measured using HPLC with multi-wavelength photodiode-array absorbance detection. 150 µL of serum was mixed with an equal volume of buffer, and then mixed with 2 volumes of ethanol containing the internal standard (tocol). The analytes were extracted from the aqueous phase into hexane. The combined hexane extracts were dried under vacuum. The extract was re-dissolved in ethyl acetate and diluted in the mobile phase. An aliquot was injected into a C18 reversed-phase column and eluted isocratically. Vitamin A was quantified by measuring the absorption at 325 nm. Serum retinol concentration ≤0.70 µmol/L was used to define vitamin A deficiency [21]. The reference method's coefficient of variation was <10%, and the assay detection limit was 0.52 µmol/L.

**Urinary iodine concentration.** Urine samples were collected in a 100ml disposable plastic screw-caped urine container. The urine samples were transported to a temporary laboratory for processing before being shipped to the central laboratory. At the temporary laboratory, the urine samples were transferred into screw-capped plastic vials and frozen at −20˚C until shipped to the Tanzania Food and Nutrition Centre Laboratory for analysis. The urine samples were analyzed using the ammonium per sulfate digestion method, as previously described by Sandell-Kolthoff reaction [22–24]. The assay's accuracy was evaluated using reference quality-control urine samples received from the CDC, Atlanta GA, USA. The reference method's coefficient of variation was <10%, and the assay detection limit was <5.0 µg/L. Median IUC for pregnant women were defined as <150 µg/L insufficient, 150–249 µg/L adequate, 250–499 µg/L above requirement and >500 µg/L excessive.

**Malaria testing.** Whole blood collected from a purple top vacutainer was used for collecting blood for a malaria test. Malaria parasitemia was identified on-site using Malaria Rapid Diagnostic Tests (MRDTs). The results were then documented on designated survey documents. Positive malaria cases were promptly informed to the study participants on the same day of testing, and refereed to the respective ANC health service provider.

## Statistical analysis

All data were uploaded to the secure server, the analysis file was created and analyzed by using Stata software version 17.0. The frequency of occurrence in percentage and median for different variables were computed. Also, chi-square test was used to determine the association between variables at 95% confidence interval (p < 0.05). Composite indicators for multiple micronutrient deficiency, trend analysis for independent variables exhibiting trends, and multi-level logistic regression were done.

## Results

### Socio-demographic information among pregnant women attended ANC in Mbeya region, 2020

There were 420 pregnant women with a mean age of 25.5 (SD 6.4) who ranged from 15 to 49 years, Table 1. More than half of respondents (55.2%) had an age range of 20–29 years, 25.2% of respondents were 30 years and above, and 19.7% of respondents were adolescents (Table 1). Most women were married or cohabiting (56.7%). The majority of women reported attaining at least a primary education (71.7%). A high proportion (88.1%, n = 370) of respondents were

**Table 1. Socio-demographic and clinical characteristics of respondents.**

| Background and Clinical characteristics | Frequency | Percent (%) |
|---|---|---|
| **Age groups (years)** | | |
| 15–19 | 83 | 19.8 |
| 20–29 | 232 | 55.2 |
| 30+ | 105 | 25.0 |
| **Marital Status** | | |
| Married / Cohabit | 238 | 56.7 |
| Single / Not Married | 182 | 43.3 |
| **Education level** | | |
| No education | 34 | 8.1 |
| Primary | 301 | 71.7 |
| Secondary and above | 85 | 20.2 |
| **Wealth Quintile** | | |
| Low | 144 | 34.3 |
| Middle | 136 | 32.4 |
| High | 140 | 33.3 |
| **Occupation** | | |
| Employed | 370 | 88.1 |
| Unemployed | 50 | 11.9 |
| **Gestational age** | | |
| Trimesters One (<12 weeks) | 110 | 26.2 |
| Trimesters Two (12–27 weeks) | 310 | 73.8 |
| **Gravidity(N = 420)** | | |
| Primigravida | 104 | 24.8 |
| Multigravida | 316 | 75.2 |
| **Antenatal Care (ANC) visit** | | |
| <2 | 164 | 39.0 |
| ≥2 | 256 | 61.0 |
| **Malaria status*** | | |
| Positive | 9 | 2.2 |
| Negative | 402 | 97.8 |
| **Anaemia status by hemoglobin level*** | | |
| Anaemic | 107 | 25.5 |
| Normal | 312 | 74.5 |
| **Iron Deficiency Anaemia (IDA)†** | | |
| IDA | 65 | 60.7 |
| Other anaemia | 42 | 39.3 |
| **Mid Upper Arm Circumference (MUAC)** | | |
| Thinness (<23cm) | 16 | 3.8 |
| Normal (≥23 cm) | 383 | 91.2 |
| | **Mean** | **SD** |
| Age (years) (N = 420) | 25.5 | 6.4 |
| Haemoglobin (g/dL)* (N = 418) | 11.7 | 1.5 |
| MUAC (Centimeters) (N = 420) | 27.2 | 3.1 |
| Ferritin Levels (μg/L)* (N = 417) | 36.7 | 48.3 |
| Alpha 1-acid Glycoprotein (AGP)* (N = 418) | 0.5 | 0.2 |
| C-Reactive Protein (CRP)* (N = 418) | 4.8 | 9.3 |
| Serum retinol (μmol/L)* (N = 418) | 1.1 | 0.4 |

(*Continued*)

**Table 1.** (Continued)

| Background and Clinical characteristics | Frequency | Percent (%) |
|---|---|---|
| Red blood cell folate (nmol/L)* (N = 418) | 1138.5 | 476.2 |
| | **Median** | **Range** |
| **Urinary Iodine Concentration**(μg/L) **(N = 419)\*** | 279.4 | 185.4–444.1 |

* Subtotals across variables might differ due to missing values

†Considered only anaemic pregnant women

Anaemia = Hb <11.0 and <10.5 g/dL for first and second trimester respectively; IDA = Anaemia accompanied with Serum Ferritin <15.0 μg/L

employed. Also, approximately three-quarters of them (73.8%) were in the second trimester (12–27 weeks), with 75.2% of respondents being pregnant at least for the second time. Most respondents (61.0%) had already visited ANC clinics at least twice. Only 2.2% of the recruited women tested positive for malaria parasites while 25.5% were found to be anaemic of which 60.7% had IDA. A high proportion of respondents (91.2%) had a mid-upper arm circumference greater than or equal to 23 cm, and the median urinary iodine among the respondents was 279.4 μg/L.

## Proportion of micronutrient deficiencies across socio-demographic characteristics of respondents

The prevalence of iron deficiency (38.4%) was the highest micronutrient deficiency, followed by folate (21.7%), vitamin B12 (9.9%), and vitamin A deficiency (9.8%), Table 2. Compared to adult aged pregnant women, adolescents had lower prevalence of each micronutrient deficiency. Except for vitamin A deficiency, highest prevalence of other micronutrients deficiencies were observed among pregnant women with no education, low wealth quantile, and employed compared to their counterpart (Table 2). Furthermore, multigravida pregnant women and those in second trimester had higher micronutrient prevalence. In contrast to pregnant women with iron and vitamin B12 deficiencies, thin pregnant women had higher vitamin A and folate deficiencies compared to normal pregnant women. Also, anaemic pregnant women had higher micronutrient deficiencies prevalence than non-anaemic women.

## Co-occurrence of micronutrient deficiencies

Single micronutrient deficiency was highly prevalent (42.8%) among pregnant women, Table 3. The majority of pregnant women with multiple micronutrient deficiency (15.0%) had double while 2.2% had triple deficiencies of any kind of micronutrient. Multiple micronutrient deficiency (MMD) was highly variable across demographic characteristics and nutritional status of pregnant women. MMD was most prevalent (20.5%) among women aged 20–29 years, and least prevalent (12.5%) in adolescents. Further, the prevalence of MMD was highest (23.5%) among women with middle wealth status as compared to other economic status. Pregnant women who reported not attaining any education had the highest prevalence of MMD (23.5%). Also, 30.2% of pregnant women who were anaemic were found to be deficient in multiple micronutrients. Nevertheless, wealth quantile, gestational age and anaemia status had a significant association ($p < 0.05$) with the occurrence of MMD among pregnant women.

Moreover, the likelihood of pregnant women encountering multiple micronutrient deficiencies escalates with advancing age (p = 0.045), Table 4. High wealth quantile was protective against MMD (p = 0.016). The women in their second trimester were more likely to have

**Table 2. Proportion of micronutrient deficiencies of respondents across background characteristics.**

|  | Vitamin A, n(%) | Iron, n(%) | *Folate, n(%) | Vitamin B12, n(%) |
|---|---|---|---|---|
| **Age group** |  |  |  |  |
| 15–19 | 7(8.6) | 27(33.3) | 12(14.6) | 8(9.9) |
| 20–29 | 25(10.8) | 96(41.7) | 52(22.5) | 23(10) |
| 30+ | 9(8.5) | 37(34.9) | 27(25.5) | 10(9.6) |
| **Marital status** |  |  |  |  |
| Married | 21(8.9) | 84(35.6) | 51(21.5) | 24(10.3) |
| Single/Not Married | 20(11) | 76(42) | 40(22.1) | 17(9.4) |
| **Education level** |  |  |  |  |
| No education | 5(14.7) | 14(41.2) | 9(26.5) | 5(14.7) |
| Primary | 29(9.7) | 113(37.8) | 67(22.3) | 27(9.1) |
| Secondary and above | 7(8.3) | 33(39.3) | 15(17.9) | 9(10.7) |
| **Occupation** |  |  |  |  |
| Employed | 39(10.6) | 140(38) | 80(21.7) | 35(9.6) |
| Unemployed | 2(4) | 20(40.8) | 11(22) | 6(12) |
| **Wealth quintiles** |  |  |  |  |
| Low | 13(9.1) | 54(38) | 37(25.7) | 20(14.1) |
| Middle | 14(10.3) | 57(41.9) | 34(25) | 15(11) |
| High | 14(10.1) | 49(35.3) | 20(14.4) | 6(4.4) |
| **Gestational age** |  |  |  |  |
| First Trimester | 4(6.7) | 12(20) | 13(21.7) | 2(3.4) |
| Second Trimester | 37(10.3) | 148(41.5)** | 78(21.7) | 39(11) |
| **Number of pregnancies experienced** |  |  |  |  |
| Primigravida | 9(8.7) | 36(34.6) | 17(16.3) | 14(13.6) |
| Multigravida | 32(10.2) | 124(39.6) | 74(23.6) | 27(8.7) |
| **ANC visits** |  |  |  |  |
| <2 | 16(9.9) | 58(35.8) | 40(24.5) | 6(3.7)** |
| ≥2 | 25(9.8) | 102(40) | 51(19.9) | 35(13.9) |
| **MUAC (Centimeters)** |  |  |  |  |
| Thinness | 3(18.8) | 4(25) | 4(25) | 1(6.3) |
| Normal | 38(9.5) | 156(38.9) | 87(21.6) | 40(10) |
| **Anaemia Status** |  |  |  |  |
| Normal | 25(8)* | 96(31)** | 65(20.8) | 31(10)*** |
| Anaemic | 16(15) | 64(59.8) | 26(24.3) | 10(9.4) |
| Total | 41(9.8) | 160(38.4) | 91(21.7) | 41(9.9) |

1 Folate insufficiency

*p<0.1

**p<0.05

***p<0.001

MMD compared to those in first trimester (p = 0.034). Also, it was noted that the likelihood of having MMD was higher among pregnant women with anaemia compared with those who were non-anaemic (p = 0.000).

## Discussion

This study presents prevalence of selected micronutrients (folate, iodine, iron, vitamin A, and vitamin B12) deficiency and provides evidence of their co-occurrence among pregnant

**Table 3. Percent of women with multiple micronutrients deficiencies across background characteristics.**

|  | Non/Single MD n(%) | Multiple MD n(%) | p-value |
|---|---|---|---|
| **Age categories(years)** | | | |
| 15–19 | 70(87.5) | 10(12.5) | 0.133 |
| 20–29 | 182(79.5) | 47(20.5) | |
| 30+ | 90(86.5) | 14(13.5) | |
| **Education level** | | | |
| No education | 26(76.5) | 8(23.5) | 0.474 |
| Primary | 248(84.1) | 47(15.9) | |
| Secondary above | 68(81) | 16(19) | |
| **Residence type** | | | |
| Rural | 237(83.2) | 48(16.8) | 0.779 |
| Urban | 105(82) | 23(18) | |
| **Marital status** | | | |
| Married | 196(84.1) | 37(15.9) | 0.422 |
| Not married/divorced | 146(81.1) | 34(18.9) | |
| **Employment Status** | | | |
| Employed | 302(83) | 62(17) | 0.816 |
| Unemployed | 40(81.6) | 9(18.4) | |
| **Wealth Quantiles** | | | |
| Low | 115(82.1) | 25(17.9) | **0.014**$^*$ |
| Middle | 104(76.5) | 32(23.5) | |
| High | 123(89.8) | 14(10.2) | |
| **Gestational age** | | | |
| First Trimester | 56(94.9) | 3(5.1) | **0.008**$^{**}$ |
| Second Trimester | 286(80.8) | 68(19.2) | |
| **Number of pregnancies had** | | | |
| Primigravida | 87(84.5) | 16(15.5) | 0.607 |
| Multigravida | 255(82.3) | 55(17.7) | |
| **Number of visits** | | | |
| <2 | 136(84) | 26(16) | 0.621 |
| 2 | 206(82.1) | 45(17.9) | |
| **MUAC (Centimeters)** | | | |
| Thinness | 15(93.8) | 1(6.3) | 0.237 |
| Normal | 327(82.4) | 70(17.6) | |
| **Anaemia Status** | | | |
| Normal | 268(87.3) | 39(12.7) | **0.000**$^{***}$ |
| Anaemic | 74(69.8) | 32(30.2) | |
| Total | 342(82.8) | 71(17.2) | |

Chi-square has been computed to test for associations between MMD occurrences with the independent variables

$^*$ p < 0.05

$^{**}$ p < 0.01

$^{***}$ p < 0.001

women in southern highland of Tanzania. Moreover, this baseline study informed the design of the implementation research project in Mbeya which aim at address the micronutrient deficiency among pregnant women.

**Table 4. Analysis of factors associated with multiple micronutrient deficiencies prevalence among pregnant women attending ANC in Tanzania.**

| Variable | PR (95%CI) | *p-value* | APR (95% CI) | *p-value* | Trend test(p-Value) |
|---|---|---|---|---|---|
| **Age Categories** | | | | | |
| 15–19 | 1 | | 1 | | |
| 20–29 | 1.68(0.84, 3.32) | 0.139 | 1.87(0.77, 4.52) | 0.167 | |
| 30+ | 1.11(0.49, 2.51) | 0.803 | 1.37(0.49, 3.88) | 0.549 | **0.045***|
| **Education Level** | | | | | |
| No formal Education | 1 | | 1 | . | |
| Primary | 0.73(0.34, 1.60) | 0.435 | 0.73(0.32, 1.66) | 0.449 | |
| Secondary and above | 0.901(0.37, 2.18) | 0.815 | 1.15(0.41, 3.19) | 0.794 | 0.225 |
| **Residence type** | | | | | |
| Rural | 1 | | 1 | | |
| Urban | 1.05(0.56, 1.98) | 0.880 | 1.15(0.68, 1.95) | 0.605 | |
| **Marital Status** | | | | | |
| Married | 1 | | 1 | | |
| Not married/divorced | 1.18(0.74, 1.88) | 0.483 | 1.27(0.77, 2.10) | 0.354 | |
| **Employment Status** | | | | | |
| Employed | 1 | | 1 | | |
| Unemployed | 1.10 (0.55, 2.22) | 0.7885 | 1.04(0.49 2.19) | 0.9163 | |
| **Wealth Quantile** | | | | | |
| Low | 1 | | 1 | | |
| Middle | 1.3(0.78 2.23) | 0.306 | 1.54(0.82, 2.56) | 0.199 | |
| High | 0.58(0.30, 1.12) | 0.105 | 0.51(0.24, 1.08) | 0.079 | **0.016***|
| **Gestational Age** | | | | | |
| First Trimester | 1 | | 1 | | |
| Second Trimester | 3.71(1.17, 11.82) | 0.026 | 4.72(1.12, 19.88) | **0.034***| |
| **Number of pregnancies** | | | | | |
| Primigravida | 1 | | 1 | | |
| Multigravida | 1.13 (0.65, 1.98) | 0.660 | 0.99(0.45, 2.19) | 0.982 | |
| **ANC Visits** | | | | | |
| <2 | 1 | | 1 | | |
| ≥2 | 1.15 (0.71, 1.88) | 0.572 | 1.12(0.65, 1.92) | 0.691 | |
| **MUAC** | | | | | |
| Thin | 1 | | 1 | | |
| Normal | 2.82(0.417, 19.00) | 0.288 | 3.19(0.398,25.585) | 0.275 | |
| **Anaemia Status** | | | | | |
| Non-anaemic | 1 | | 1 | . | |
| Anaemic | 2.92(1.78, 4.80) | 0.000*| 2.61(1.57, 4.34) | **0.000**** | |

Logistic regression and trend analysis have been computed to test for likelihood of MMD occurrence against the independent variables

PR, Prevalence ratio; APR, Adjusted Prevalence Ratio

* p < 0.05

** p < 0.01

The overall prevalence of anaemia in this study was 25.5%; a moderate public health concern [25]. The prevalence was lower than national anaemia prevalence (56%) [8]; similar to other studies conducted in different parts of the country [26, 27]. Lower anaemia prevalence compared to national prevalence might be attributed to differences in modes of blood collection. There is evidence that hemoglobin measurement precision varies between venous and

capillary blood samples, leading to variability in hemoglobin estimates [28]. However, based on the prevalence from this survey and that of national survey, more initiatives are required to further reduce the prevalence to the lower levels, from mild to no public health concern. Micronutrient deficiencies among pregnant women were also established; ID prevailed the highest (38.4%) followed by folate (21.7%), vitamin B12 (9.9%) and vitamin A (9.8%). The high prevalence of ID might be caused by absorption complexity of dietary iron based on the form of iron consumed; non-heme iron requires absorption enhancers such as vitamin C [29]. Since iron has a big role during haemoglobin formation, anaemia is attributed to iron deficiency (ID) approximately 50% [25]. This was also evident in this current study as more than half of the anemic pregnant women had iron deficiency (Table 2). Also, vitamin A deficiency prevalence among anaemic pregnant women was almost twice (15%) of that among non-anaemic pregnant women (8%). This might be due to the vitamin A role in facilitating iron metabolism within the body [30]. In contrast to other micronutrients deficiencies, iodine deficiency was not a problem among pregnant women. The median urinary iodine concentration was above 150µg/L indicating adequate iodine intake as per WHO cut-off points [31]. Adequate intake of iodine might be attributed to the high coverage of universal salt iodization program in the country [8].

The evidence of micronutrient deficiency co-occurrence among pregnant women by 17.2% was established. The likelihood of pregnant women encountering MMD escalated with advancing age in the present study (p = 0.045). Further, adolescent pregnant women had lower micronutrients deficiency prevalence's compared to adult pregnant women. This may be attributed to the number of pregnancies and the spacing between children, as multigravida pregnant women also exhibited higher levels of micronutrient deficiency. This aligns with the conclusion of the certain study, stated that multigravidity is a risk factor for anaemia in pregnancy [32]. Women who are pregnant for the subsequent time may be at a higher risk of micronutrient deficiencies, as they need to support the nutritional needs of both the current pregnancy and any existing children. Nevertheless, the double burden of micronutrients demand during adolescence resulting from growth spurt and pregnancy might have also contributed to the total prevalence of MMD among pregnant women [33].

Wealth quantile showed a significant association (p = 0.014) with MMD occurrence. This might be due to the significant association between wealth index and dietary diversity practice during pregnancy [34]. Mbeya region is among regions with high food crop production in the country together with livestock activities within the region [35]. Even though FAO has reported the improved trend of animal-source foods (ASF) consumption over the years in sub-Saharan Africa, their consumption is still a challenge mostly in developing countries due to its cost [6]. In comparison to ASF, low cost of many plant-based foods lead majority of populations in developing countries to rely much on them which consequently causes deficiencies of some micronutrients. Inability of a pregnant woman to afford animal based foods limits some nutrient intake and may affect the quality of nutrients taken [36]. ASF are the major source of heme iron (the readily absorbable form of iron), retinol (the active form of vitamin A), primary food source for vitamin B12 and partly a source of folate [37, 38]. Through changes in public policies that raise household incomes or lower ASF prices there is a potential to increase the ASF consumption that contributes to improved food and nutrition security [39]. Moreover, the higher the economic status in combination with nutrition awareness the better access and use to diversify foods among pregnant women within the region, the less MMD occurrence. Even though it was not significantly associated, pregnant women with no education had higher chances to experience MMD (23.5%). The ANC guideline instructs the health service providers to conduct nutrition counselling among pregnant women during each ANC contact [40]. Educated pregnant women might be more likely to quickly understand during nutrition counselling at ANC visits and practice due to critical thinking capacity.

The more the pregnancy grows the more energy and nutrients requirements increase to support both maternal and child well-being for good pregnancy outcome. Physiological increase in maternal blood volume and accelerated fetal development in the final phase of pregnancy causes diverse micronutrient deficits to be more frequent as a woman approaches the last quarter of pregnancy [41]. This was evidently in this study as findings revealed that gestational age had significant association with MMD occurrence (p = 0.008). Also, pregnant women in their second trimester were 3.5 more likely to experience multiple micronutrients deficiency compared to their counterparts (p = 0.034). Hence, continued inadequate micronutrients intake during pregnancy may escalate micronutrients deficiency with advancing gestational age.

The co-occurrence of micronutrient deficiencies among pregnant women might have contributed to anaemia onset causing poor pregnancy outcomes. This is due to the fact that anaemia etiology is multifactorial, with nutrition being a major cause especially attributed to ID, but also other micronutrients deficiencies including folate, vitamin A and B12 [25]. Each of these micronutrients has a role to play either in iron metabolism or blood formation. Effect of their deficiencies to cause anaemia is evidently in this study as 30.2% of pregnant women who were anaemic also had MMD. Also, the two had a strong significant association (p < 0.000). Hence, ensuring iron, folate, vitamin A and B12 adequacy during pregnancy can contribute to the reduction of anaemia prevalence.

In Tanzania, there is a national program of routine iron and folic acid supplementation (IFA) among pregnant women as per WHO guidelines. In TDHS 2022 it was reported that 81% of pregnant women took any iron-containing supplements, but adherence to daily intake remains questionable as high prevalence of anaemia still exists. Factors such as history of abortion, nutrition education awareness, proper service provision during ANC visits, and residing in urban have been reported to influence IFA intake adherence in developing countries [42, 43]. Barriers to IFA adherence commonly reported in developing countries include bad taste, nausea, constipation, dark colour of the stool, vomiting, long administration time and unavailability of the supplements [44]. Introducing daily multiple micronutrient supplementations (MMS) among pregnant women could overcome barriers to IFA adherence while providing a broader range of micronutrients. MMS contain 15 essential nutrients to support a healthy pregnancy, including vitamin A, D, E, B1, B2, B3, B6, B12, folic acid, vitamin C, elemental iron, zinc, copper, selenium and iodine. These supplements are designed to address a range of micronutrient deficiencies in populations, particularly vulnerable groups such as pregnant women [45].

Despite the robust findings, this study has several limitations. It only included pregnant women attending ANC clinics at selected health facilities, missing those in the broader community. The cross-sectional design restricts the ability to establish causality between observed deficiencies and socio-demographic factors; longitudinal studies are needed for a deeper exploration of these relationships. Additionally, the study's focus on the Mbeya region suggests that future research should cover other regions for a more comprehensive national perspective.

## Conclusions

This study underscores the critical issue of MMD among pregnant women in Mbeya, Tanzania. The high prevalence and co-occurrence of these deficiencies indicates the need for a renewed strategy to improve maternal nutrition. Strategies should include supporting and encouraging adequate nutrition among pregnant women by enhancing access to micronutrients through practice dietary diversity. Increasing access to MMS and fortified foods can also

be helpful, as these are proven measures for reducing the burden of micronutrient deficiencies. Additionally, the study advocates for establishing nutrition biomarker surveillance systems to track the health and nutrition status of vulnerable groups like pregnant women. These biomarkers reflect not only the intake but also the metabolism of nutrients and possibly, effects on health outcomes.

## Acknowledgments

We extend our sincere appreciation to the pregnant women who participated in the study, health facilities' administration and healthcare professionals, local government authorities in Mbeya region, and everyone else who contributed to the accomplishment of this survey.

## Author Contributions

**Conceptualization:** Geofrey Mchau, Hope Masanja, Abraham Sanga, Ramadhani Mwiru, Charity Zvandaziva, Ramadhan Noor, Ray Masumo, Germana Leyna.

**Data curation:** Geofrey Mchau, Hope Masanja, Erick Killel, Kaunara Azizi, Tedson Lukindo, Adam Hancy, Stanislaus Henry, Heavenlight Paul, Kudakweshi Chimanya, Abela Twinomujuni.

**Investigation:** Geofrey Mchau, Hope Masanja.

**Methodology:** Geofrey Mchau, Hope Masanja, Stanislaus Henry, Ray Masumo, Germana Leyna.

**Resources:** Abraham Sanga, Ramadhani Mwiru, Ramadhan Noor, Patrick Codjia.

**Supervision:** Geofrey Mchau, Ray Masumo.

**Writing – original draft:** Geofrey Mchau, Hope Masanja, Erick Killel, Kaunara Azizi, Stanislaus Henry, Heavenlight Paul.

**Writing – review & editing:** Geofrey Mchau, Hope Masanja, Erick Killel, Kaunara Azizi, Tedson Lukindo, Adam Hancy, Stanislaus Henry, Heavenlight Paul, Abraham Sanga, Ramadhani Mwiru, Charity Zvandaziva, Kudakweshi Chimanya, Abela Twinomujuni, Ramadhan Noor, Ray Masumo, Germana Leyna, Patrick Codjia.

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
