## [Decision Letter · Decision Letter 0]

21 May 2024

PONE-D-24-06850Co-occurrence of micronutrient deficiencies among pregnant women in Mbeya region of Tanzania: A call for a renewed approach to improve maternal nutritionPLOS ONE

Dear Dr. Masanja,

Thank you for submitting your manuscript to PLOS ONE. After careful consideration, we feel that it has merit but does not fully meet PLOS ONE’s publication criteria as it currently stands. Therefore, we invite you to submit a revised version of the manuscript that addresses the points raised during the review process.

We look forward to receiving your revised manuscript.

Kind regards,

Jai K Das

Academic Editor

PLOS ONE

Reviewers' comments:

Reviewer's Responses to Questions

**Comments to the Author**

1. Is the manuscript technically sound, and do the data support the conclusions?

Reviewer #1: Yes

Reviewer #2: Partly

Reviewer #3: Yes

2. Has the statistical analysis been performed appropriately and rigorously? 

Reviewer #1: No

Reviewer #2: No

Reviewer #3: I Don't Know

3. Have the authors made all data underlying the findings in their manuscript fully available?

Reviewer #1: No

Reviewer #2: Yes

Reviewer #3: Yes

4. Is the manuscript presented in an intelligible fashion and written in standard English?

Reviewer #1: Yes

Reviewer #2: Yes

Reviewer #3: Yes

5. Review Comments to the Author

Reviewer #1: Overall, this is a very important study that highlights the prevalence of multiple micronutrient deficiencies among pregnant women in Tanzania. The methods are sound, and the interpretation of results not inappropriate, however, adding additional information will improve the manuscript considerably.

My major concern is related to the analysis of the data to accurately reflect the sampling strategy. Given that 44 health facilities were randomly selected, and that pregnant women attending those facilities were also randomly selected, the estimates of micronutrient deficiencies could be considered representative of the region. The calculation of estimates needs to take into account the correlated nature of women attending the same ANC clinic. It would be similar to a cluster-randomized design, where women (or households) in the same cluster have measured or unmeasured similarities, and therefore estimates need to have a confidence interval or somehow include an estimate of the variance (e.g., standard error to accompany the mean). Even the sample size determination section (line 114) states that there was a design effect of 1.5, which implies that there was clustering, but the statistical analysis and presentation of results do not include information about controlling for that correlation at the cluster (or health facility level). If the authors can resolve that issue and make a few modifications to their manuscript, then this paper will hold more authority in the future as it will be referenced for years to come.

My second major concern is related to consistency within the laboratory methods section. For vitamin B12 and urinary iodine, there is discussion of QC procedures. Would it be possible to describe if QA or QC was done for all analytes, and include the results from QC consistently when it was done (e.g., report on the CV for B12 as was done for UI)?

My third major concern is the use of an appropriate or accurate cutoff to define anemia for pregnant women. They have cutoffs that differ from non-pregnant women, and it seems that the cutoff for non-pregnant women was used incorrectly here (line 160 - "Hemoglobin levels <12.0 ... were used to characterize anaemia")

Minor concerns:

Introduction section – please include information on what current policies or programs for pregnant women exist in Mbeya? and cite any evaluation of those programs (if they exist, and if they don't then list as an important evidence gap).

Sample size – was it powered to detect any subgroup differences, or just produce reliable point estimates, and if the latter – of what, iron deficiency, etc.

Laboratory sample analysis section:

(1) Line 149 - RBC folate concentration < 748 nmol/L to indicate deficiency or “insufficiency”?

(2) Line 154 – When reporting QC then please include information about range of CV from those runs.

(3) Please add references for all cutoffs (e.g., B12 <148 pmol/L). This tool may be helpful in identifying the original sources for cutoffs to add as references: https://mnsurvey.nutritionintl.org/phases/19

(4) Why use Hb < 12.0 g/L to define anemia among pregnant women? They have a unique cutoff that is trimmest specific, and it seems that you would have that level of detail. Please modify the analysis using the correct trimester-specific cutoff to define low hemoglobin for pregnant women (https://www.who.int/publications/i/item/9789240088542)

(5) Line 164 – The BRINDA “adjustment” for inflammation uses a regression approach, which is different from a correction factor approach. Thus, the language around interpretation of ferritin for iron deficiency is confusing and needs to be clarified.

(6) Lines 185 – 187: The assay’s accuracy was evaluated using reference quality-control urine samples received from the CDC, Atlanta GA, USA. The reference method's coefficient of variation was <10%, and the assay detection limit was <5.0 μg/L. It is great to have this level of detail for UI but please be consistent across all micronutrient biomarkers and if QA or QC was not performed indicate that rather than leave the readers guessing. (cite EQUIP here, by adding website)

(7) Was treatment provided to positive malaria cases? If so, what was it? And was there any additional follow-up? Beyond SOC or same as SOC?

Results – line 214 states "25% were found to be anaemic of which 60.8% had IDA". In addition to needing to revise all estimates for anemia, the abstract would need to be modified because as it is currently written it seems like IDA was found among 60% of pregnant women -- it does not classify among those with anemia 60% had IDA - and too many people will only read an abstract, which has an incorrect statement.

Table 1 – ANC visit total is higher than 420 (typo?)

- Age group, add unit: years,

- Where is the category cohabitating? Because that was described in text but does not appear in the table?

- Add information (into the methods) on how wealth was defined

- Add detail on how gestational age was determined (last menstrual period or ultrasound, or a combination)

- Define any abbreviation in the footnotes (e.g., ANC, or just spell it out)

- Include cutoffs in the footnote for how anemia was defined, and how IDA was defined.

- If you had information on inflammation, then it would be good to include in table 1 also

- You need to revise all estimates with robust standard errors accounting for correlation among women recruited from the same facility

Table 2 – Please add statistical testing to determine if differences in deficiencies were statistically significant by characteristics (e.g., age, marital status, etc.), similar to how it was done in table 3. It would not require additional columns, but starring to suggest where differences were detected. Consider Rao-Scott modified chi square to account for clustering.

Line 238 – be consistent in terminology – here “insufficiency” is listed rather than deficiency, but these are two different things.

Line 274 – the difference in anemia prevalence could have been partially based on mode of blood collection (venous vs. capillary) for the study vs. national estimates. That needs to be discussed here please. Consider this recent work [Assessing Accuracy and Precision of Hemoglobin Determination in Venous, Capillary Pool, and Single-Drop Capillary Blood Specimens Using 3 Different HemoCue Hb Models: The Multicountry Hemoglobin Measurement (HEME) Study - PubMed (nih.gov)]

Nowhere do you mention that the mUIC is in the range of "250-499 Above requirements" based on the most recent guidelines for assessment of urinary iodine (available here: https://www.unicef.org/nutrition/files/Monitoring-of-Salt-Iodization.pdf)

Reviewer #2: The study is well written and the methodology is clear and well explained. However, conclusions are being made in the analyis without presenting the CI of the results which could change the conclusions being taken in case the C.I. overlaps. Therefore, I would recommend to add the C.I. to all prevalence results.

PR and APR in Table 4 are acronyms that are not explained before.

I do not see limitations in the discussion session, could they be added ? For example that the results only reflects the population of pregnant women coming to ANC.

Some conclusions on the recommended programmes : fortified foods and MMS lack justification as to why those 2 are the most adequate. Indeed, we might want to also have more sustainable solution.

Reviewer #3: This article reports on the prevalence of micronutrient deficiencies in pregnant women attending ANC in Tanzania and was conducted to inform a research project. The introduction, justification and tables in the results section could be clearer. It might be worth adding a multivariate analysis to better understand the predictors of the micronutrient deficiencies.

My specific comments are as follows.

Title:

Co-occurrence of micronutrient deficiencies among pregnant women in Mbeya region

of Tanzania: A call for a renewed approach to improve maternal nutrition

The title sounds like an advocacy document but it is actually a cross-sectional study so perhaps the title can be slightly adjusted?

Abstract:

The numbers in the abstract and text don’t match. For example in the abstract triple deficiencies is 18% while in the main text it is 2%.

Introduction:

The introduction could be tightened up a bit and repetition removed. E.g. the fact that micronturitents deficiencies lead to adverse outcomes is mentioned 3 times (line 54-56, line 63 and 71/72).

Line 53 – 63: there is a bit of repetition, perhaps this could be tightened up a bit.

Line 75 onwards: Can this paragraph be clearer. As someone who is not involved with interventions for pregnant women I could use a bit more information here. In general can you be clearer on what the current strategy is? Is IFAS just recommended in the context of research? Why only in the context of research? Does this mean that they are currently not routinely provided to pregnant women? Maybe you can clarify if MMS are currently provided to pregnant women or not? If they are not provided is this because of lack of evidence or lack of access? Or are MMS a potential alternative strategy to be used instead of IFA?

Line 83 “Additionally, the results will guide the design of implementation research for MMS among pregnant women, laying the basis for increased access to MMS in Tanzania”

This could be clearer. Is the implementation research looking at effectiveness of MMS compared to a different supplement, whether supplements are needed or what the barriers are to access?

Line 84. Full stop missing

Methods:

Line 100: You talk about probability proportional to size. Size of what? The number of patients enrolled in ANC, catchment area or typical case load?

Line 108: More details on how the systematic random sampling was done? Or was it simple random sampling?

Please also state how many women were invited to participate in the study at each facility? Was it an equal number at each facility?

Line 163: It might be worth clarifying that ferritin is an acute phase protein and hence the adjustment. Was the adjustment done for all or only those with inflammation? Was inflammation measured? Can you describe the method used for correction. Are these the correction factors developed by Thurnham etal or does Brinda use a different apporach. This information is needed to be able to interpret the iron deficiency data.

I don’t see information on Ethical approval and consent procedures mentioned in the manuscript. This would be useful to add.

Results:

Of the women that were invited to participate in the study, how many declined to participate? This should be stated in the results.

In the abstract it says that 60% if pregnant women had IDA, however in Table 1, it is stated that only 65 women had IDA which is 15% of 420.

The prevalence estimates should be accompanied by confidence intervals.

Table 2: The table is a bit difficult to read overall? Perhaps it would help adding in the numbers for the groups mentioned in the first column. It would also help to see if there are any significant differences between the groups and which test was used to determine associations.

Table 3. Why are no and single in one category? Would it not make sense to have multiple categories (none, one and multiple)

Table 4. What does PR and APR stand for.

General: Consider adding multivariate regression analysis to better understand the predictors.

Discussion

Line 271-273: Could you provide more information on the implementation research project? Perhaps also in the introduction to make this clearer?

Is the article trying to determine whether IFAS should potentially be replaced by MMS, and this what the implementation research will look at?

Line 295-298: “Further, adolescent pregnant women had lower micronutrients

deficiency prevalence’s compared to adult pregnant women. This may be attributed to the number of pregnancies and the spacing between children, as multigravida pregnant women also exhibited higher levels of micronutrient deficiency.” What happens if you adjust the regression model for number of pregnancies to confirm this statement. In fact in your analysis you could consider both the univariate and multivariate models?

Line 332, what are these “counterparts”? are they pregnant women in the first trimester?

Line 350, can you be clearer what these MMS supplements are, which nutrients do they consist of? How are they different from IFA? And are IFA currently routinely given to pregnant women?

Do you know if this sample is representative of the population? i.e. do the majority of woman access ANC? Could the prevalence in the community be even higher?

Are there any limitation of this study?

Conclusion

I am not sure I fully follow the last sentence, what biomarker of disease processes are you referring too?

6. PLOS authors have the option to publish the peer review history of their article (what does this mean?). If published, this will include your full peer review and any attached files.

Reviewer #1: No

Reviewer #2: No

Reviewer #3: No

---

## [Author Response · Author response to Decision Letter 0]

25 Jul 2024

Editor 

- The whole manuscript has been restructured to meet PLOS ONE style 

2. Please include your full ethics statement in the ‘Methods’ section of your manuscript file. In your statement, please include the full name of the IRB or ethics committee who approved or waived your study, as well as whether or not you obtained informed written or verbal consent. If consent was waived for your study, please include this information in your statement as well 

- Ethics statement has been added under the method section 

Reviewer 1 

1. Major concern 1: My major concern is related to the analysis of the data to accurately reflect the sampling strategy. Given that 44 health facilities were randomly selected, and that pregnant women attending those facilities were also randomly selected, the estimates of micronutrient deficiencies could be considered representative of the region. The calculation of estimates needs to take into account the correlated nature of women attending the same ANC clinic. It would be similar to a cluster-randomized design, where women (or households) in the same cluster have measured or unmeasured similarities, and therefore estimates need to have a confidence interval or somehow include an estimate of the variance (e.g., standard error to accompany the mean). Even the sample size determination section (line 114) states that there was a design effect of 1.5, which implies that there was clustering, but the statistical analysis and presentation of results do not include information about controlling for that correlation at the cluster (or health facility level). If the authors can resolve that issue and make a few modifications to their manuscript, then this paper will hold more authority in the future as it will be referenced for years to come. 

- Thank you for the observation. It is true that the sampling procedure employed cluster-design with health facilities (cluster) being sampling points. We have re-analyzed the data; women are now nested within council in the multi-level logistic regression model. 

2. Major concern 2: For vitamin B12 and urinary iodine, there is discussion of QC procedures. Would it be possible to describe if QA or QC was done for all analytes, and include the results from QC consistently when it was done (e.g., report on the CV for B12 as was done for UI)?

- The reference method's coefficient of variation and the assay detection limit has been added to all micronutrient biomarkers.

3. Major concern 3: They have cutoffs that differ from non-pregnant women, and it seems that the cutoff for non-pregnant women was used incorrectly here (line 160 - "Hemoglobin levels <12.0 ... were used to characterize anaemia 

- Thank you for point out this. This was an overlook during writing and not during data analysis. We appreciate you mentioning it. The statement has been corrected and cited appropriately. 

4. (1)Line 149 - RBC folate concentration < 748 nmol/L to indicate deficiency or “insufficiency”? The use of the term insufficiency for folate has been maintained due to it’s significance. Assessing folate insufficiency ensure both the mother and fetus have adequate folate levels for optimal health and development, rather than deficiency which indicates critically low folate levels. 

- The use of the term insufficiency for folate has been maintained due to it’s significance. Assessing folate insufficiency ensure both the mother and fetus have adequate folate levels for optimal health and development, rather than deficiency which indicates critically low folate levels

5. (2) Line 154 – When reporting QC then please include information about range of CV from those runs. 

- All sections that requires QC has been revised 

6. (3) Please add references for all cutoffs (e.g., B12 <148 pmol/L). This tool may be helpful in identifying the original sources for cutoffs to add as references: https://mnsurvey.nutritionintl.org/phases/19

- The citation was added for all cutoffs; vitamin B12, Thank you for the link, it is very useful. 

7. (4) Why use Hb < 12.0 g/L to define anemia among pregnant women? They have a unique cutoff that is trimmest specific, and it seems that you would have that level of detail. Please modify the analysis using the correct trimester-specific cutoff to define low hemoglobin for pregnant women (https://www.who.int/publications/i/item/9789240088542)

- This was an overlook during writing and not during data analysis, and it has been corrected. “Hemoglobin levels <11.0 and <10.5 g/dL were used to characterize anaemia for pregnant women in the first and second trimester respectively. Hemoglobin levels <7.0 g/dL was used to characterize severe anaemia. Serum Ferritin < 15.0 g/L was used to identify iron deficiency. Hb level <11 g/dL with Serum Ferritin <15.0 g/L was used to define iron deficiency anaemia.” 

8. (5) Line 164 – The BRINDA “adjustment” for inflammation uses a regression approach, which is different from a correction factor approach. Thus, the language around interpretation of ferritin for iron deficiency is confusing and needs to be clarified. 

- Sentence has been rephrased

9. (6) Lines 185 – 187: The assay’s accuracy was evaluated using reference quality-control urine samples received from the CDC, Atlanta GA, USA. The reference method's coefficient of variation was <10%, and the assay detection limit was <5.0 μg/L. It is great to have this level of detail for UI but please be consistent across all micronutrient biomarkers and if QA or QC was not performed indicate that rather than leave the readers guessing. (cite EQUIP here, by adding website) 

- The reference method's coefficient of variation and the assay detection limit has been added to all micronutrient biomarkers. 

10. (7) Was treatment provided to positive malaria cases? If so, what was it? And was there any additional follow-up? Beyond SOC or same as SOC? 

- The research team was not responsible for providing treatment to identified malaria cases. Instead, the malaria cases were referred to the ANC health service provider on the respective clinic for treatment. 

11. Line 214 states "25% were found to be anaemic of which 60.8% had IDA". In addition to needing to revise all estimates for anemia, the abstract would need to be modified because as it is currently written it seems like IDA was found among 60% of pregnant women -- it does not classify among those with anemia 60% had IDA - and too many people will only read an abstract, which has an incorrect statement.

- The statement in the abstract has been corrected to explain 60.7% (approximated to 61%) IDA was among anaemic pregnant women.

12. Table 1 

- ANC visit total is higher than 420 (typo?); 

Total number of ANC visits has been corrected, it was a typo

- Age group, add unit: years,

“years” has been added as unit under age group

- Where is the category cohabitating? Because that was described in text but does not appear in the table?

Cohabit has been added under married category, now reading as “married/cohabit”. It was an overlook, thanks. 

- Add information (into the methods) on how wealth was defined

Information describing wealth quantile has been added in methods

- Add detail on how gestational age was determined (last menstrual period or ultrasound, or a combination)

Information on gestation age determination has been added under methods. It was direct read from ANC card which normally determined by referring to the client’s last menstrual period. Ultrasound checks for gestation age are rare, since are used as the second option in case the first method is not reliable

- Define any abbreviation in the footnotes (e.g., ANC, or just spell it out)

ANC has been spelled out “Antenatal Care (ANC)”

- Include cutoffs in the footnote for how anemia was defined, and how IDA was defined.

Cutoffs for anaemia and IDA have been added at the footnote.

- If you had information on inflammation, then it would be good to include in table 1 also

Information on CRP and AGP have been included in Table 1

- You need to revise all estimates with robust standard errors accounting for correlation among women recruited from the same facility - 

Thank you for the observation. This is similar to what was proposed earlier, as a major comment. We have accounted for similarities within councils using a multi-level regression.

13. Table 2 – Please add statistical testing to determine if differences in deficiencies were statistically significant by characteristics (e.g., age, marital status, etc.), similar to how it was done in table 3. It would not require additional columns, but starring to suggest where differences were detected. Consider Rao-Scott modified chi square to account for clustering. 

- Re-analyzed data, stars inserted. 

14. Line 238 – be consistent in terminology – here “insufficiency” is listed rather than deficiency, but these are two different things.

- Corrected, insufficiency has been replaced with deficiency 

15. Line 274 – the difference in anemia prevalence could have been partially based on mode of blood collection (venous vs. capillary) for the study vs. national estimates. That needs to be discussed here please. Consider this recent work [Assessing Accuracy and Precision of Hemoglobin Determination in Venous, Capillary Pool, and Single-Drop Capillary Blood Specimens Using 3 Different HemoCue Hb Models: The Multicountry Hemoglobin Measurement (HEME) Study - PubMed (nih.gov)] 

- The section has been revised 

16. Nowhere do you mention that the mUIC is in the range of "250-499 Above requirements" based on the most recent guidelines for assessment of urinary iodine (available here: https://www.unicef.org/nutrition/files/Monitoring-of-Salt-Iodization.pdf) 

- The section has been revised 

Reviewer 2 

1. Conclusions are being made in the analyis without presenting the CI of the results which could change the conclusions being taken in case the C.I. overlaps. Therefore, I would recommend to add the C.I. to all prevalence results. 

- Confidence intervals added in table of model estimates

2. PR and APR in Table 4 are acronyms that are not explained before. 

- PR and APR has been defined in a footnote. PR, Prevalence ratio; APR, Adjusted Prevalence Ratio

3. I do not see limitations in the discussion session, could they be added? For example that the results only reflects the population of pregnant women coming to ANC. 

- Limitation has been added under the discussion. It covers the recruitment of pregnant women only who visited ANC clinics, the cross-sectional nature of the study and inclusion of only Mbeya region.

4. Some conclusions on the recommended programmes: fortified foods and MMS lack justification as to why those 2 are the most adequate. Indeed, we might want to also have more sustainable solution. 

- Improving dietary diversity have been added as recommended strategies to intervene the problem. However, fortified foods and MMS has also been retained as they are evidence-based, cost-effective, and can significantly improve the nutritional status of pregnant women, particularly in settings with high prevalence of multiple micronutrient deficiencies.

Reviewer 3 

1. Title: Co-occurrence of micronutrient deficiencies among pregnant women in Mbeya regionof Tanzania: A call for a renewed approach to improve maternal nutrition. 

- The title sounds like an advocacy document but it is actually a cross-sectional study so perhaps the title can be slightly adjusted? Thanks for point out this, and the title has been modified to “Micronutrient Deficiencies and their Co-occurrence among Pregnant Women in Mbeya Region, Tanzania”

2. Abstract: The numbers in the abstract and text don’t match. For example in the abstract triple deficiencies is 18% while in the main text it is 2%. 

- Has been corrected to adopt that in the texts

3. Introduction: The introduction could be tightened up a bit and repetition removed. E.g. the fact that micronturitents deficiencies lead to adverse outcomes is mentioned 3 times (line 54-56, line 63 and 71/72). Line 53 – 63: there is a bit of repetition, perhaps this could be tightened up a bit. Line 75 onwards: Can this paragraph be clearer. As someone who is not involved with interventions for pregnant women I could use a bit more information here. In general can you be clearer on what the current strategy is? Is IFAS just recommended in the context of research? Why only in the context of research? Does this mean that they are currently not routinely provided to pregnant women? Maybe you can clarify if MMS are currently provided to pregnant women or not? If they are not provided is this because of lack of evidence or lack of access? Or are MMS a potential alternative strategy to be used instead of IFA? 

- The introduction has been rephrased; repetitions has been removed, current strategies been elaborated more, as well as current practice of IFA VS MMS has been elaborated. 

4. Line 83 “Additionally, the results will guide the design of implementation research for MMS among pregnant women, laying the basis for increased access to MMS in Tanzania”. This could be clearer. Is the implementation research looking at effectiveness of MMS compared to a different supplement, whether supplements are needed or what the barriers are to access? 

- The implementation research will not evaluate the effectiveness of MMS, as this has already been established in previous studies. Instead, it will focus on assessing the acceptability and adherence of MMS compared to the currently used IFA supplementation, as well as barriers to supplementation adherence.

5. Line 84. Full stop missing

- Full stop has been added, 

6. Line 100: You talk about probability proportional to size. Size of what? The number of patients enrolled in ANC, catchment area or typical case load? 

- The section has been revised, and missing explanation about the population to which probability proportional to size was applied has been added.

7. Line 108: More details on how the systematic random sampling was done? Or was it simple random sampling? 

- Rephrased to indicate that the number of selected facilities in each district was determined by the total number of facilities in from each respective district. Further, the number of women to enroll in the study was also determined prior the survey by having a list of pregnant women who were scheduled for attendance and calculating sampling interval K. Selection was done systematically with replacement after every Kth pregnant woman.

8. Please also state how many women were invited to participate in the study at each facility? Was it an equal number at each facility?

- Total number of women interviewed in each health facility was proportionally distributed depending on the total number of pregnant women attending ANC in the specific facility. Since it was based on proportions, different number of participants was recruited in each health facility. 

9. Line 163: It might be worth clarifying that ferritin is an acute phase protein and hence the adjustment. Was the adjustment done for all or only those with inflammation? Was inflammation measured? Can you describe the method used for correction. Are these the correction factors developed by Thurnham etal or does Brinda use a different apporach. This information is needed to be able to interpret the iron deficiency data. 

- Ferritin was adjusted by using regression modal as applied in BRINDA project.

10. I don’t see information on Ethical approval and consent procedures mentioned in the manuscript. This would be useful to add. 

- Ethical statement has been added

11. Results: Of the women that were invited to participate in the study, how many declined to participate? This should be stated in the results. 

- Refusal rate has been incorporated in the manuscript. 

12. In the abstract it says that 60% if pregnant women had IDA, however in Table 1, it is stated that only 65 women had IDA which is 15% of 420. 

- It was elaborated on the footnote of Table 1 that, the value for IDA considered only anaemic pregnant women. Meaning, out of 107 anaemic pregnant women 65 (61%) had IDA. The 60% at the abstract has also been c

---

## [Decision Letter · Decision Letter 1]

15 Aug 2024

Micronutrient deficiencies and their co-occurrence among pregnant women in Mbeya region, Tanzania

PONE-D-24-06850R1

Dear Dr. Masanja,

We’re pleased to inform you that your manuscript has been judged scientifically suitable for publication and will be formally accepted for publication once it meets all outstanding technical requirements.

Kind regards,

Jai K Das

Academic Editor

PLOS ONE

Reviewers' comments:

Reviewer's Responses to Questions

**Comments to the Author**

1. If the authors have adequately addressed your comments raised in a previous round of review and you feel that this manuscript is now acceptable for publication, you may indicate that here to bypass the “Comments to the Author” section, enter your conflict of interest statement in the “Confidential to Editor” section, and submit your "Accept" recommendation.

Reviewer #1: All comments have been addressed

Reviewer #2: All comments have been addressed

2. Is the manuscript technically sound, and do the data support the conclusions?

Reviewer #1: Yes

Reviewer #2: Yes

3. Has the statistical analysis been performed appropriately and rigorously? 

Reviewer #1: Yes

Reviewer #2: Yes

4. Have the authors made all data underlying the findings in their manuscript fully available?

Reviewer #1: Yes

Reviewer #2: Yes

5. Is the manuscript presented in an intelligible fashion and written in standard English?

Reviewer #1: Yes

Reviewer #2: Yes

6. Review Comments to the Author

Reviewer #1: Good job responding to reviewer comments. Nesting women within council and reanalysis of the data, adding QA and QC (it seems that primarily QC data were added) and confirming the Hb cutoffs make this a much stronger contribution.

Reviewer #2: The comments made by the authors were adressed thoroughly and allow more robustness from the paper, limitations are clear and fit the recommendations.

7. PLOS authors have the option to publish the peer review history of their article (what does this mean?). If published, this will include your full peer review and any attached files.

Reviewer #1: No

Reviewer #2: No
